# Characterizing the Roles of Life Stage and Season on the Prevalence of Select Viral Pathogens in *Acheta domesticus* Crickets on a Commercial Cricket Farm in the United States

**DOI:** 10.3390/vetsci12030191

**Published:** 2025-02-20

**Authors:** Kimberly L. Boykin, Amy Bitter, Zoey N. Lex, John Tuminello, Mark A. Mitchell

**Affiliations:** Department of Veterinary Clinical Sciences, Louisiana State University, Skip Bertman Dr, Baton Rouge, LA 70803, USA

**Keywords:** *Acheta domesticus* densovirus, *Acheta domesticus* volvovirus, invertebrate iridovirus 6, cricket viruses, insects as feed and food, mass-reared insects, epidemiology, entomopathogenic virus

## Abstract

Millions of crickets are grown annually for the purposes of human and animal consumption. Crickets are an excellent source of nutrition and can be grown under much more sustainable conditions than traditional livestock species. However, the cricket industry has been plagued by disease outbreaks that have impacted their production numbers. To better understand these outbreaks, samples were collected across different age groups and seasons, and tested for the presence of three industry-important viruses. Two of the viruses, *Acheta domesticus* densovirus and invertebrate iridovirus 6, were more prominent in the colder winter months, indicating that environmental factors like temperature and humidity may play a role in viral transmission and replication. These viruses also showed either higher prevalence rates or viral loads as the crickets aged, which corresponds to the higher death rates seen in older cricket stocks. The third virus, *Acheta domesticus* volvovirus, maintained high prevalence rates year-round and across age groups. Understanding how these viruses spread is instrumental in knowing how best to manage commercial populations of crickets.

## 1. Introduction

The insects-as-feed-and-food industry has steadily grown in popularity over the course of the last decade, with more and more western cultures becoming accepting of the idea of using insects as a source of nutrients for both animals and people [1,2]. Increases in global temperatures, greenhouse gases, deforestation, and overfishing are major drivers for the shift to using mini-livestock (i.e., insects) over more traditional sources of animal protein [1,2]. Insects require much less feed, water, and land to grow comparable amounts of useable protein, and are considered the most economical and sustainable way to raise a source of animal protein [1,2]. Unfortunately, certain sectors of the insect industry have been plagued by disease outbreaks that have impacted the supply of insects, and thus decreased the total economic revenue generated from this industry [3]. Similar to other agricultural markets, high animal densities and management practices put insects at high risk for infectious diseases, with viral pathogens being the most concerning [3].

*Acheta domesticus*, the European house cricket, is one of the most popular insects grown as part of the edible insect industry, but it has suffered from several worldwide viral outbreaks since the 1970s [4,5,6,7,8]. The most severe outbreaks in the USA and Europe have resulted in 100% mortality rates and have completely bankrupted some cricket operations or forced the producers to switch to other less desirable species of crickets, only to be plagued by new viral epidemics. To date, viruses from the *Parvoviridae*, *Iridoviridae*, and *Dicistroviridae* families have been associated with most large-scale epidemics [3,9]. Recent work with next-generation sequencing has revealed the presence of several other viruses; however, only a fraction of those have been characterized [10]. For those that have been characterized, clinical signs are often non-specific, making it difficult to distinguish which viral pathogen is associated with a morbidity or mortality event, and, in some cases, more than one virus may be impacting a population. Viral-infected crickets often experience a slower growth rate and decreased life span, and those reaching adulthood have been shown to produce fewer eggs [11,12]. Prevalence, transmission, and virulence studies are currently lacking for most cricket viruses; therefore, the main goal of this study was to determine the prevalence for three of the viruses known to infect *A. domesticus* on commercial farms within the United States, and to determine if the life stage or season played a role in disease transmission. If the cricket industry is going to be viable and provide animal protein to growing populations of humans and animals, it is vital that we develop a better understanding of the epidemiology of these viruses.

Knowing more about how a virus is transmitted and what factors play a role in its transmission is fundamentally important when making decisions relating to cricket management. For viruses that are horizontally transmitted, reducing the spread of the virus to uninfected stock is of vital importance. Bins, water and food dishes, and the tools used to manage the insects should be thoroughly disinfected after each use. Younger insects should be grown separately from older stocks (and ideally have separate personnel) to limit the spread to naïve populations. For vertically transmitted viruses, where spread cannot be as easily limited, emphasis may be placed more on breeding for resistance or genetic manipulation. Other changes could be made to ensure less stress and healthier immune systems, such as focusing on high-quality nutrition, the addition of probiotics to feed, and smaller stocking densities. Environmental factors like temperature and humidity are also known to affect the transmission of some viruses. Cricket paralysis virus, a member of the *Dicistroviridae* family, was shown to have increased virion formation at 30 °C as compared to 25 °C, but was inhibited again at temperatures of 37 °C [13]. The western tent caterpillar (*Malacosoma californicum pluviale*) died more readily under laboratory conditions from a baculovirus infection as the temperature increased (up to 30 °C), but the opposite was found under field conditions due to the behavioral changes in the larvae [14]. Therefore, studying how the prevalence may change across life stages and season/temperature may impact how we currently manage these insects in a mass-rearing facility.

A single large-scale commercial cricket farm in the southern United States was chosen as our principal site for cricket collection. Preliminary studies had already confirmed the presence of several viruses, including *Acheta domesticus* densovirus (AdDV), *Acheta domesticus* volvovirus (AdVVV), and invertebrate iridovirus type 6 or cricket iridovirus (CrIV), on the farm using either conventional PCR or next-generation sequencing techniques. Using this farm as a model for other infected farms across the country, we hoped to gain further insight into the epidemiology of these viruses, focusing on the overall prevalence, degree of co-infections, and possible risk factors contributing to these diseases. Anecdotal reports suggested mortality rates were at their highest in later-stage animals (>4 weeks old) and during the winter months, when the building temperatures were coolest. Based on our pilot data, we hypothesized that the viral prevalence and/or viral load would be higher in 4-to-6-week-old crickets compared to 2-week-old crickets, and that the overall prevalence for each of the three viruses would be ≥45% in these older cricket stocks. We also hypothesized that the prevalence of each virus would be the highest during October and January compared to the warmer months of May and August. This study represents the first attempt to systematically characterize the prevalence of any of these viruses and their potential risk factors (age and season) at a cricket-rearing facility in the United States. Characterizing the prevalence and potential risk factors associated with these viruses will enable individuals in the industry to develop biosecurity and management methods to minimize the impact of these viruses in the future.

## 2. Materials and Methods

### 2.1. Cricket Selection

A cross-sectional study was performed to estimate the prevalences of AdDV, AdVVV, and CrIV in *A. domesticus* at a single large-scale cricket farm in the southern United States. The following three different life stages of crickets were sampled: 2-week-old (nymphs ¼″ in size), 4-week-old (nymphs ½″ in size), and 6-week-olds (adults 1″ in size). Nymphs and adults were distributed between five different buildings, and the buildings contained multiple life stages at any one time (mixed cohorts). At least five batches from each available life stage were collected from each building at each sampling point. A random number generator (random.org, Accessed on 30 October 2019) was used to determine which of the available cricket bins within a building was sampled. A batch consisted of 15 crickets from a single bin that housed approximately 4000 crickets each. All of the bins were 3′ × 5′ × 2.5′ (0.91 × 1.5 × 0.76 m; W × L × D) plastic, open-aired bins. Each sampling period consisted of a minimum of 20 cricket batches for a single age group. The sample size selected for this study was based on the following a priori data: an alpha = 0.05, a power = 0.8, and an expected viral prevalence of >45% in the later-stage groups. The crickets were sampled at the following four different time periods: October 2019, January 2020, May 2020, and August 2020. The delay in our Spring 2020 sampling period was due to COVID-19 restrictions. All of the samples were collected using sterile methods (sterile gloves [Sensicare PI, Medline Industries, Inc., Northfield, IL, USA] and whirl-pak bags [Nasco Sampling LLC, Pleasant Prairie, WI, USA]) to prevent cross-contamination between the bins. Crickets were transported on wet ice to the Louisiana State University School of Veterinary Medicine, where they were immediately frozen and stored at −80 °C until further processing.

### 2.2. Temperature and Humidity Data

To assess the roles of temperature and humidity on the prevalence of cricket viruses, indoor and outdoor temperatures and humidity data were collected for the two weeks prior to each collection date. Indoor data were collected using combined temperature and humidity digital data loggers (EL-USB-2, Lascar Electronics, Erie, PA, USA) that recorded readings once every hour. Loggers were positioned in the center of a room/building at least 10 feet away from a heater exhaust vent, outer wall, or doorway. The average, minimum, and maximum outdoor temperatures and humidities were recorded using local data from weatherunderground.com (Accessed on 1 October 2020) and timeanddate.com/weather (Accessed on 1 October 2020).

### 2.3. DNA Extractions

DNA was extracted using a Qiagen DNeasy Blood and Tissue kit (Qiagen, Hilden, Germany). All of the samples from a batch were prepped as individual crickets or as smaller pools (in the case of 2-week-old crickets) to reach a minimum of 50 mg of tissue. In the 6-week-old crickets, the gastrointestinal tract, Malpighian tubules, hypodermis, and fat body were dissected out and used for extractions to reduce the amount of tissue used to <200 mg. Each individual cricket (or pool of crickets) was crushed using a mortar and pestle immediately after being removed from the −80 °C freezer. The cricket homogenates were then mixed with 180 µL Buffer ATL and 20 µL proteinase K, and incubated at 56 °C overnight. An amount of 200 µL of AL buffer was added to the sample, vortexed, and incubated for an additional 10 min. An amount of 200 µL of 100% ethanol was added to the sample and vortexed again. Samples were then transferred to spin columns, centrifuged, and washed twice (with AW1 and AW2 buffers, respectively). DNA was eluted in 200 µL of AE buffer. The DNA quality and concentration for each sample was estimated using a Thermo NanoDrop ND-1000 UV/VIS Spectrophotometer (ThermoFisher Scientific, Waltham, MA, USA). Due to the extensive number of samples (>4000) and the higher cost associated with quantitative PCR (qPCR), the individual extractions from a batch (15 crickets) were pooled together (2 µL of each sample) for use as the template DNA for qPCR. The remaining individual extraction material was frozen at −80 °C for additional future projects.

### 2.4. Viral qPCR

Quantitative real-time PCR assays were designed de novo for AdDV and AdVVV. The assay for CrIV used primers and probes that had already been published [15]. The primers and probes used for AdDV were located on the first non-structural protein (NS) and corresponded to a 100 bp segment (GenBank identification: NC_075118.1): forward (5′-GATTATCTGCCGTACCGTTCT-3′), probe (5′-TGGATACGGAATACCGAACT GATGAC-3′, and reverse (5′-TCAGTACCCAATCA TGTACCAC-3′). The primers and probes used for AdVVV were located within the overlapping second and third open reading frames and corresponded to a 113 bp segment (GenBank identification: KC794539.1): forward (5′-CGAGTCCCTTTCTGAGGTTTC-3′), probe (5′-CAGGGCTCCAGATG TGCTGATTGA-3′), and reverse (5′-CCTCCAATTCTGCATTTCCTTTAC-3′). The PCR reactions were run using TaqMan^®^ Universal PCR Master Mix and a QuantStudio 12K Flex Real-Time PCR System (Applied Biosystems, Foster City, CA, USA). Constructed plasmids that included the corresponding DNA sequences and nuclease-free water were used as positive and negative controls, respectively, for each virus. For AdDV and AdVVV, the reaction was set up to include 400 nM of forward and reverse primers each, 250 nM of probe, and 2 μL of DNA template in a 10 μL reaction. For CrIV, the reaction was set up to include 900 nM of forward and reverse primers each, 250 nM of probe, and 1 μL of DNA template in a 10 μL reaction. The cycling conditions for AdDV and AdVVV were as follows: (1) 98 °C for 2 minutes to activate the enzyme; (2) 40 cycles at 98 °C for 10 seconds to allow for denaturation, and 58 °C at 30 s each for annealing and extension. The cycling conditions for CrIV were as follows: (1) 50 °C for 2 minutes; (2) 40 cycles at 95 °C for 15 seconds, and 65 °C at 60 s each. All of the assays were run in triplicate. A Ct value of ≥35.000 was used as the cutoff for consideration as a negative test. Due to the large number of samples, four plates were used for each virus, corresponding to each month of sampling. A set of test samples (one from each sampling month) was included on every plate to assess the interplate variability. An interplate coefficient of variation was calculated from each of the test sample’s Ct values for each of the viruses (AdDV: 2.9–5.8%; AdVVV: 0.3–1.1%; CrIV: 1.1–3.6%). A standard curve was included on every plate, and the curve efficiencies ranged from 92.5 to 101.7%.

### 2.5. Statistical Analysis

Univariate testing (Chi Square or Fisher’s exact test) was used to determine if there were any differences in the viral prevalence between the three age groups, four seasons, or five buildings. Independent variables with a *p* < 0.20 at the univariate level were used to construct a binary logistic regression model to determine the odds ratios and 95% confidence intervals for each of the significant independent factors. A stepwise approach was used for the model building. The Hosmer–Lemeshow goodness-of-fit test was used to assess the model fit. Biologically relevant interaction terms were also included in the model. Ordinal logistic regression was used to construct models evaluating the estimated viral loads as the dependent variable, and age, building, season, and temperature/humidity data as independent variables. To determine the significance of viral co-infections, a Spearman’s rank-order correlation was constructed between each of the virus pairings using either the estimated viral prevalence (ordinal) and/or the prevalence data (categorical). All of the statistical analyses were performed using SPSS 28.0 software (IBM Statistics, Armonk, NY, USA). A *p* < 0.05 was considered significant.

## 3. Results

### 3.1. Temperature and Humidity Data

The outdoor temperature and humidity data are presented in Figure 1. As expected, January was the coldest sampling month in terms of lowest minimum (−0.5 °C, 31 °F;), daily average (12.9 °C, 55.2 °F), and maximum temperatures (26.1 °C, 79 °F), followed by October (min: 1.7 °C, 35 °F; avg daily: 17.4 °C, 63.3 °F; max: 30 °C, 86 °F). May and August were much warmer and had much less variance between days (May min: 15 °C, 59 °F, avg daily: 24.6 °C, 76.2 °F, and max: 32.8 °C, 91 °F; Aug min: 20.6 °C, 69 °F, avg daily: 27.2 °C, 81 °F, and max: 35 °C, 95 °F). The daily relative humidity varied from day to day throughout the year, but the average for each month remained at 75–85%.

Indoor temperature and humidity data are presented in Table 1, with a breakdown for each building. On average, Buildings 1 and 2 stayed warmer (above 32.2 °C, 90 °F) than Buildings 4, 5, and 6; however, during August, only Building 6 maintained a daily average temperature below 32.2 °C (<90 °F), as the outdoor temperatures were much warmer during this month. For humidity, January and October saw lower indoor humidities (both 45%) than in May (65.2%) and August (74.2%).

### 3.2. Viral Prevalence Data

AdDV—The overall prevalence of AdDV was 46.7%. The positivity rates steadily increased from the 2- (37%, 95% CI: 30–48%), 4- (48.1%, 95% CI: 38–58%), to 6-week-old crickets (55.9%, 95% CI: 46–65%) (Table 2). By season, January had the highest prevalence, at 75.3% (95% CI: 66–85%), followed by October (59.2%, 95% CI: 47–71%), August (34%, 95% CI: 24–44%), and May (22.1%, 95% CI: 13–31%) (Table 2). Table 3 shows positivity rates for each month divided out by building, with Building 2 having the highest positivity rate overall (64.4%, 95% CI: 50–79%).

Using univariate statistics, prevalence rates by age, season, and building were all statistically significant (age: χ^2^ = 8.3, *p* = 0.015; season: χ^2^ = 59.7, *p* < 0.001; building: χ^2^ = 20.3, *p* < 0.001); therefore, a binary logistical regression model was constructed to better determine the odds ratios for each factor. Age, season, and building were the only variables included in the final model. A Hosmer and Lemeshow goodness-of-fit test was performed to assess the fit of the model (*p* = 0.207) and indicated a good performance. For age, 4- and 6-week-old crickets had an odds ratio of 2.1 (95% CI: 1.1–3.9) and 2.76 (95% CI: 1.5–5.1), respectively, compared to 2-week-old crickets (Wald = 10.8, *p* = 0.004). This indicates that older crickets were more than twice as likely to test positive than younger nymphs. For buildings, Building 1 was determined to be the most protective building (Wald = 12.1, *p* = 0.016), with Building 2 (OR: 3.8; 95% CI: 1.6–8.7) being the least protective. For season, samples in January (OR: 6.4; 95% CI:3.2–13) and October (OR: 2.9; 95% CI: 1.4–5.8) were significantly more likely to be positive compared to the reference month of August (Wald = 46.2, *p* < 0.001). A forest plot graphically displaying the odds ratios is included in the Appendix A.

We also looked at changes in the estimated viral loads for each of our factors (see Figure 2 and Figure 3). An ordinal logistic regression was constructed with the best-fitting model including age, building, and season as factors, and building average temperatures as a covariate (model fit: χ^2^ = 84.391, *p* < 0.001). Following a similar trend to the positivity rates, 2-week-old (Wald = 15.5, *p* < 0.001; odds ratio = 0.3; 95% CI = 0.189–0.571) and 4-week-old crickets (Wald = 6.2, *p* = 0.013; odds ratio = 0.5; 95% CI: 0.3–0.9) had lower viral loads compared to 6-week-old crickets. This result was likely driven by a large spike in viral loads amongst 6-week-old crickets in August, despite that month having lower prevalence rates than January and October (Figure 2). Most of that spike appears to have occurred in Building 5 (Figure 3), but this building was not significantly different from Building 6, which was used as the reference building in the model (Building 5: Wald = 1.8, *p* = 0.173). Building 1 was the only building that produced a significant result when compared to the reference building and appeared to have a protective effect with lower viral loads associated with that building (Wald = 4.041, *p* = 0.044; odds ratio = 0.35; 95% CI: 0.13–0.97).

As for month, the model did not follow trends that we could easily visualize in the two figures, but did correlate well with what we found for the prevalence rates. Using August as the model’s reference point, January and October both had increased odds of having higher viral loads (Jan: Wald = 19.4, *p* < 0.001, odds ratio = 4, and 95% CI: 2.1–7.3; Oct: Wald = 5.4, *p* = 0.02, odds ratio = 2.1, and 95% CI: 1.1–4), while May continued to have the lowest chances of having a higher viral load (May: Wald = 4.9, *p* = 0.026; odds ratio = 0.45; 95% CI: 0.23–0.90). A forest plot graphically displaying the odds ratios is included in the Appendix A.

AdVVV—The overall prevalence of *Acheta domesticus* volvovirus was 100%. Due to this, we were unable to perform univariate statistics or a binary logistic regression on the positivity rates (infected: yes/no), as all of the crickets tested positive for the virus. Therefore, an ordinal logistic regression model was constructed using the estimated viral load (how much virus was present in the samples, 10^n^ virus particles) as the dependent variable, and the factors age, season, and building as independent variables (model fit: χ^2^ = 471.9, *p* < 0.001). Estimated viral loads are presented in Figure 4 and Figure 5.

Two-week-old crickets were found to be 4.2 times more likely to have higher viral loads as compared to six-week-old crickets (Wald = 17.3, *p* < 0.001; 95% CI = 2.1–8.4). Four-week-old and six-week-old crickets were not significantly different from each other. There was a large spike in the viral load amongst the 2-week-old crickets during the month of August (Figure 4), as well as a minor spike in the viral loads for weeks 4 and 6 during the month of August, all of which were highly significant (Oct: Wald = 3117.5, *p* < 0.001, odds ratio = 0, and 95% CI = 0–0; Jan: Wald = 3058.6, *p* < 0.001, odds ratio = 0, and 95% CI = 0–0; May: odds ratio = 0 and 95% CI = 0–0). All of the buildings were impacted by this spike (Figure 5), with Building 5 seeing the largest impact; however, this was not significant (Wald = 0.4, *p* = 0.386; Building 6 used as the reference). A forest plot graphically displaying the odds ratios is included in the Appendix A.

CrIV—Similar to AdVVV, CrIV also had an overall prevalence of 100%. An ordinal logistic regression model was constructed using the estimated viral load (number of virus particles, 10^n^) as the dependent variable, and the factors age, season, and building as independent variables (model fit: χ^2^ = 206.6, *p* < 0.001). Estimated viral loads are presented in Figure 6 and Figure 7. For age, we see that younger crickets tend to have the lowest viral loads overall (Figure 6) (2-week-old: Wald = 23.6, *p* < 0.001, odds ratio = 0.33, and 95% CI = 0.17–0.47; 4-week-old: Wald = 15.8, *p* < 0.001, odds ratio = 0.36, and 95% CI = 0.21–0.59; 6-week-old: used as the model reference).

For season, we see that loads are highest for all age groups during the month of January (Wald = 139.7, *p* < 0.001). Crickets in January were 63.6 (95% CI = 31.9–126.7) times more likely to have higher viral loads when compared to the model’s reference month of August, which saw the smallest viral loads out of any month. October and May were 6.2 (95% CI = 3.4–11.6) and 13.9 (95% CI = 7.6–25.6) times more likely to have higher loads than August, respectively (Oct: Wald = 33.86, *p* < 0.001; May: Wald = 72.6, *p* < 0.001).

As for building, the constructed model suggested that Buildings 1 and 2 were significantly different (Building 1: Wald = 17.1, *p* < 0.001, odds ratio = 0.3, 95% and CI = 0.17–0.53; Building 2: Wald = 5.50, *p* = 0.019, odds ratio = 0.45, and 95% CI = 0.23–88) as compared to the reference of Building 6. A forest plot graphically displaying the odds ratios is included in the Appendix A.

### 3.3. Viral Co-Infections

The viral loads of each virus were analyzed to determine if a higher viral load of one virus correlated to a higher or lower prevalence or viral load of another virus. Using a Spearman rank-order correlation, we found that the viral loads of AdVVV and CrIV were moderately negatively correlated (rho = −0.49, *p* < 0.001), and that AdVVV was also mildly negatively correlated with AdDV (rho = −0.25, *p* < 0.001). No other significant associations were found related to co-infections.

## 4. Discussion

Based on the higher mortality rates seen at this farm in the colder winter months, we had hypothesized that warmer building temperatures may have a protective effect against one or more of the viruses. From our indoor temperature data, we were able to confirm that average and minimum building temperatures were at their lowest during the month of January, followed by the month of October (average building temperatures—Oct: 32.6 °C, Jan: 32.5 °C, May: 32.7 °C, and Aug 33.3 °C; minimum building temperatures—Oct: 25 °C, Jan: 25 °C, May: 26.5 °C, and Aug: 28.5 °C). This corresponds well with the seasonal prevalence rates of AdDV and the seasonal viral loads of CrIV, indicating that these two viruses may play a role in the increased winter mortalities seen at this farm. However, the temperature deviations between the seasons were minor, with only a 0.2 °C difference in the average temperature between the months of January and May, which saw drastically different prevalence rates and viral loads. However, when we add in the humidity data, a stronger case can be made for what may be driving these seasonal changes. For the average and minimum relative humidities, we see much larger deviations between the months, with January and October again having the lowest values (average relative humidity—Oct: 41.6%, Jan: 46.2%, May: 65.2%, and Aug: 73.0%; minimum relative humidity—Oct: 21.5%, Jan: 16%, May: 41%, and Aug: 49.5%). Although neither of these factors were significant in our logistic regression models for any of the viruses, we believe there is enough evidence to support further microclimate testing of both temperature and humidity in laboratory-controlled conditions to determine how the viruses and the crickets′ immune systems respond to various environmental conditions.

Both temperature and humidity are known to have an impact on the viral replication and transmission rates of other viruses. For endothermic animals, we see this quite readily with the fever response. Certain immune mediators reset the hypothalamus to induce higher body temperatures, which, in effect, can lower viral replication by way of decreased entry into host cells, the alteration of genome transcription, and the improvement of host defense mechanisms against certain viruses [16]. For ectothermic animals that are not capable of producing an internally controlled fever response, they can move to and utilize higher environmental temperatures, known as a behavioral fever, to see the same benefits [17]. Multiple insect studies, including those performed in crickets, have shown similar results. Adamo and Lovett howed that *Gryllus texensis* crickets kept at higher temperatures (33 °C) increased the activity of two immune-related enzymes and had enhanced resistance to the bacteria *Serratia marcescens* [18]. These crickets also saw increased egg-laying, faster egg development, and greater weight gains. Cevallos and Sarnow found that higher temperatures could protect insect cells from infection by cricket paralysis virus. Although higher temperatures did result in increases of viral RNA and proteins, virion formation, and, therefore, infectivity, were decreased at the highest tested temperature of 37 °C [13]. A recent study has also been released regarding temperature and AdDV. Through use of qPCR, Takacs et al. found that viral replication was the highest at 25 °C and the lowest at 35 °C; however, this result was not always repeatable throughout all of the individual experimental runs. Humidity for that study was maintained at 75–85%, which, from our results, may not be representative of real-world conditions, and may be difficult to maintain in large commercial facilities [19].

There are numerous studies related to the influenza virus and how cold temperatures paired with low humidity are associated with the seasonal outbreaks we see with that virus [20,21,22]. Influenza’s viral stability in an aerosolized form has been shown to be the most stable at low humidities (20–40% RH), low at intermediate humidity (50%), and increase again at higher humidities (60–80% RH), showing a bimodal distribution [23]. Unfortunately, due to the lack of transmission studies, we do not know if AdDV or CrIV can be spread through aerosolization. One study [12] did find evidence of AdDV DNA within air filters at an affected farm, which could suggest spread through aerosols. If that is the case, and AdDV aerosols also have a bimodal distribution, then this could help explain why August saw higher prevalence and viral loads as compared to May, when the humidity was not as high within the buildings. Exposure of the host to low humidity has been found to be detrimental as well. In mice, low-humidity conditions impaired mucociliary clearance and tissue repair functions, making them more susceptible to influenza [24]. Although insects lack a mucociliary clearance mechanism, they are still at risk of dehydration in conditions of low humidity [25]. The effect of dehydration on the immune system of insects has been understudied, but likely leads to increased stress, an altered metabolism, and an immunocompromised state [26]. As stated previously, the authors believe that further in-depth study of the microclimate of cricket-rearing facilities is needed to further explore the roles of temperature and humidity in the epidemiology of AdDV and CrIV; however, these preliminary real-world results suggest that these environmental parameters do play a role in the epidemiology of these viruses.

For AdVVV, the highest viral loads were seen in August, suggesting the opposite of what was found for AdDV and CrIV, that higher temperatures and humidities may lead to increased viral replication for this virus. Given that cricket survival was supposedly higher in the summer months at this farm, this could indicate that AdVVV is not as pathogenic as the other two viruses, or that, like cricket paralysis virus, it only undergoes increased viral replication under these conditions, increasing the viral DNA that is found via qPCR, but the transmission/infectivity of the virus is hampered during this time. Further transmission studies need to be performed to determine what is happening at the cellular level for this virus.

Life stage was another important factor that we wanted to investigate with this study. Anecdotal information from the producers, and previous papers regarding AdDV, suggested that the clinical signs of viral infection (e.g., paralysis and decreased appetite) did not start to appear in the crickets until they reached about four weeks of age [8,12,27]. We wanted to determine if crickets of all ages had similar prevalence rates and viral loads but required several weeks to start developing signs of infection, or if they gradually increased in viral load over time until they reached a point where infection switched from covert to overt in nature. The prevalence rates of AdDV did increase as the crickets aged, suggesting that not all of the crickets were exposed to the virus at a young age, or even transovarially. Four- and six-week-old crickets were more than two times as likely to be infected than two-week-old crickets. Knowing that horizontal transmission is an important part of this virus’s epidemiology can be used to make management decisions, such as separating younger and older cricket stock to reduce the spread of the virus, which would hopefully translate to improved yields. When looking at the estimated viral loads across each month (Figure 2), there was no definitive increase in the titer as the crickets aged until the month of August was considered. The large spike in 6-week-old viral loads in August is likely the driving factor behind the ordinal logistic regression model showing a significantly increased likelihood of high viral loads in older crickets. What led to this spike in August is still currently unknown but could be related to higher temperatures improving the viral replication rates (but not necessarily infectivity rates), as was seen with cricket paralysis virus [13] and potentially with AdVVV.

For AdVVV, two-week-old crickets were most likely to have higher viral loads. This was driven by a large spike that occurred in the month of August within a single building (Building 5). While all age groups in August had higher loads, it is unlikely that increased viral replication due to higher temperatures would disproportionately affect the youngest age group. A more likely explanation could be that the farm received a shipment of eggs or younger crickets from another cricket facility with higher viral loads around the same time as our sampling, which falsely increased the loads that we were seeing. Due to changes in consumer demand, unexpected mortalities, or just wanting to increase genetic variation within the breeding stock, it is not an uncommon practice for farms to sell extra crickets to other farms. This practice is also probably one of the largest contributors to why we believe most cricket farms are currently infected with multiple viruses, and why the full eradication of these viruses is unlikely to occur.

Viral loads for CrIV appear to follow the pattern of increasing as the crickets age, with two- and four-week-old crickets having lower odds ratios than six-week-old crickets. Given that this virus had a 100% prevalence rate and the load increased as the crickets aged, there is a high chance that this virus is spread vertically, which has also been found in other insect groups with iridovirus [28,29]. If it is spread vertically, this limits a producer’s ability to control the spread to younger generations through cohort separation and disinfection. Management plans would instead need to focus on improving cricket genetics and immune function, providing good nutrition and probiotic support, and limiting stress as much as possible.

When defining the epidemiology of a virus, it is important to also characterize the pathogen in relation to other infectious diseases. Because there were at least three viruses at this facility, it was important to understand whether co-infections can lead to higher prevalence rates or viral loads. At this facility, higher viral loads of AdVVV were associated with a lower viral load of CrIV and a lower prevalence of AdDV. However, as mentioned previously, two-week-old crickets had the highest viral loads of AdVVV due to the spike seen in August, and the other two viruses both had their lowest prevalence rates and viral loads in this age group. Therefore, the results of our correlation are likely misleading, and no true association exists between these viruses.

There were several limitations to the design of this study. The first was lack of control over environmental conditions. The best way to determine whether temperature or humidity influence viral prevalence or rates would be to conduct lab-controlled studies. We had no control over unseasonably warm or cold outdoor temperatures and limited control over the heating and humidification of the buildings on the farm. While this methodology was not ideal for the true determination of the temperature and humidity, that was not the purpose of this study. Our primary goal was to determine the prevalence of these viruses under real-world, large-scale industrial conditions. As was mentioned earlier with western tent caterpillars and baculovirus, mortality was increased in warmer temperatures when reared under laboratory conditions, but, due to changes in behavior in a field setting, the opposite was found to be true, and warmer temperatures were more protective against the virus. While lab-controlled studies serve an important purpose, measuring prevalence rates under field conditions was necessary to determine what factors are the most important for further study. The relatively low viral loads of some of the viruses was also a potential limitation. Most of the positive samples for AdDV and AdVVV were 10^1^–10^3^, which corresponded to Ct values close to our plate cutoff of 35. The true determination of the positive and negative status and the true titer level is challenging because the PCR has a higher variability between replicates and becomes more stochastic in nature with the presence of less template [30]. A more sensitive laboratory technique, such as digital PCR, may be needed to determine the true positivity rates. Additionally, the use of viral RNA quantification to indicate active infection could be used; however, given the extremely high prevalence rates in this study, which increased our confidence in calling these active infections, this line of testing was not pursued. The use of pooled samples (to increase the amount of DNA available for extraction in the smaller crickets and to reduce the number of individual samples that were analyzed with qPCR) limited our ability to measure the prevalence to the level of the individual cricket, but we were required to do so to ensure that we had sufficient DNA for testing, and because of financial constraints on the number of samples that could be analyzed. Regardless, because the crickets are group-housed, sampling by bin did provide insight into the risk of being positive under standard commercial conditions. Finally, we did not have real-time morbidity and mortality rates for the study; however, these are not consistently collected by the commercial operation. Future studies should work to correlate the viral findings over time with the morbidity and mortality rates.

## 5. Conclusions

Viral diseases have been linked to costly disease epidemics in captive-bred *Acheta domesticus* populations worldwide. The high mortality rates and decreased fecundity associated with these diseases have put a massive economic strain on the cricket industry and decreased our ability to integrate this sustainable animal protein source into more feed applications. To date, these economic costs have not been quantified, but will be necessary to determine the extent of these losses and to reinforce the need for establishing preventive control measures to reduce these losses, similar to other livestock species. This study was the first attempt to systematically characterize the prevalence of several viruses on a large-scale cricket farm in the United States so as to help identify preliminary risk factors that could be used to design longitudinal experimental studies in the future. From our collected data, we can confirm that lab-controlled temperature and humidity studies should be performed on each of these viruses, because each one displayed some level of seasonality to either prevalence rates, viral load, or both. Having more detailed information related to viral replication and transmission at various environmental conditions would be beneficial to help producers determine the ideal rearing conditions that crickets should be kept at to minimize the impact of these diseases. We also believe that it would be beneficial to determine if any of these viruses can be transmitted vertically or through aerosolization, as those factors could impact how best to separate and house crickets to limit the spread of disease between different populations [3,31,32,33].

## Figures and Tables

**Figure 1 vetsci-12-00191-f001:**
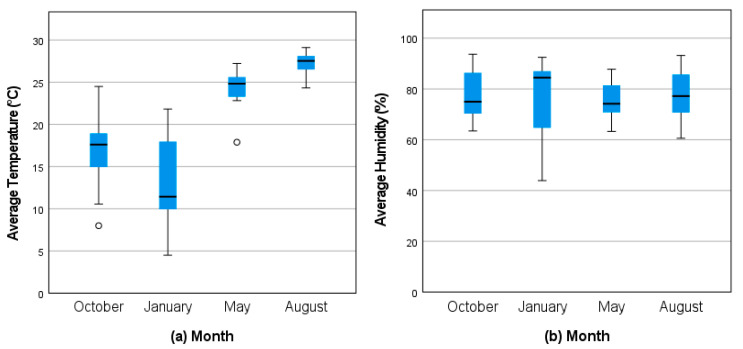
(**a**) Outdoor temperatures and (**b**) relative humidities by month.

**Figure 2 vetsci-12-00191-f002:**
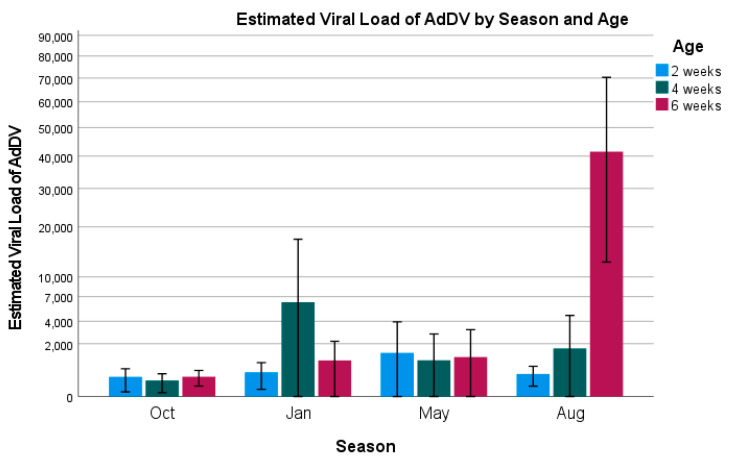
Estimated viral load of AdDV by season and age. Error bars indicate ± 2 standard error.

**Figure 3 vetsci-12-00191-f003:**
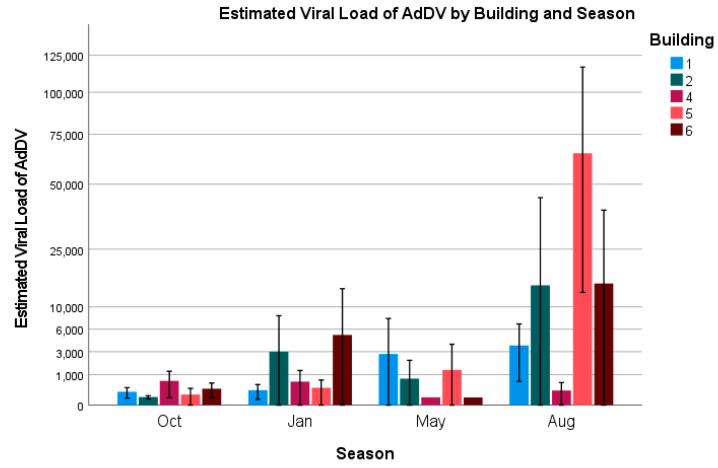
Estimated viral load of AdDV by building and season. Error bars indicate ± 2 standard errors.

**Figure 4 vetsci-12-00191-f004:**
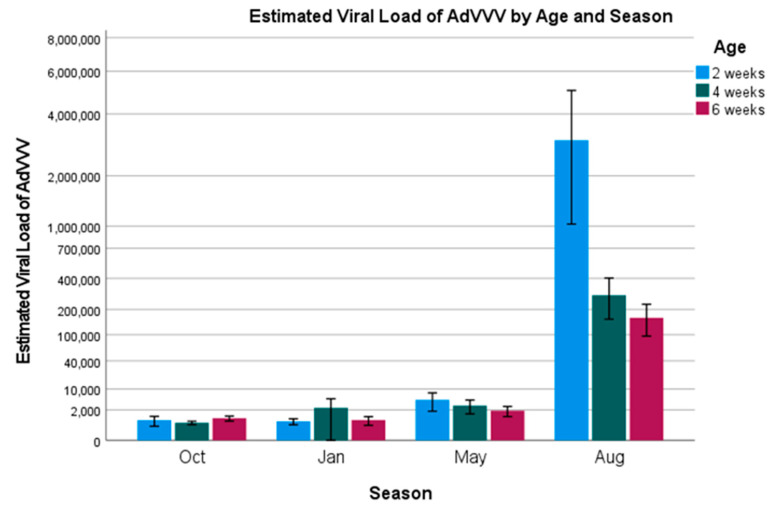
Estimated viral load of AdVVV by age and season. Error bars indicate ± 2 standard errors.

**Figure 5 vetsci-12-00191-f005:**
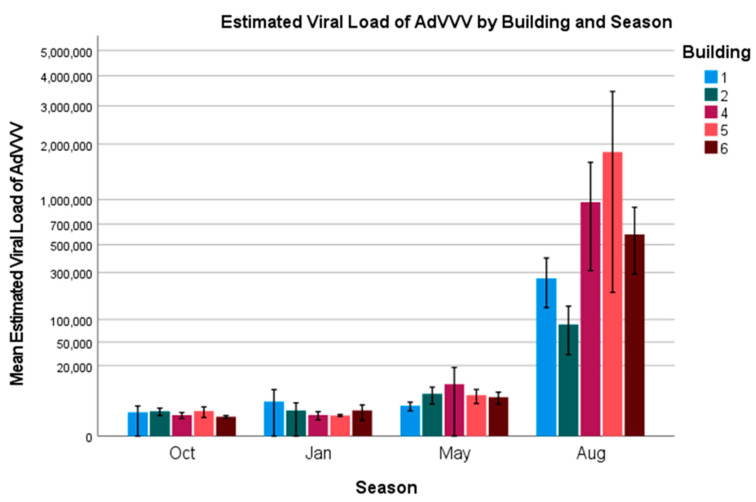
Estimated viral load of AdVVV by building and season. Error bars indicate ± 2 standard errors.

**Figure 6 vetsci-12-00191-f006:**
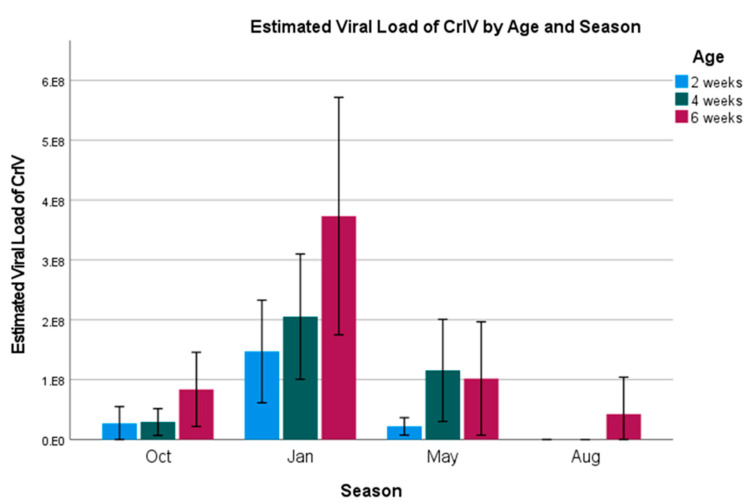
Estimated viral load of CrIV by age and season. Error bars indicate ± 2 standard errors.

**Figure 7 vetsci-12-00191-f007:**
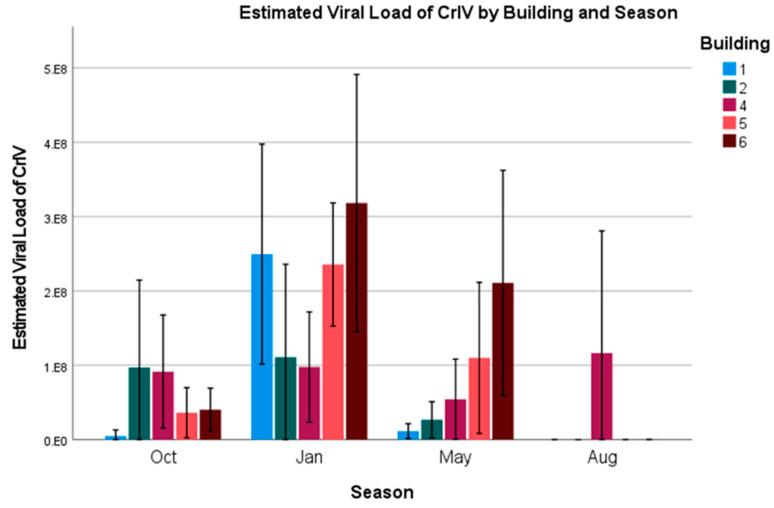
Estimated viral load of CrIV by building and season. Error bars indicate ± 2 standard errors.

**Table 1 vetsci-12-00191-t001:** Indoor temperatures and relative humidities by sampling month and building.

	Minimum Temperature and Humidity			Maximum Temperature and Humidity			Average Temperature and Humidity		
**October**	**°C**	**°F**	**RH%**	**°C**	**°F**	**RH%**	**°C**	**°F**	**RH%**
Building 1	34	93.2	23	40	104	59	35.7	96.3	40
Building 2	28	82.4	32	35.5	95.9	62.5	33.8	92.8	42.7
Building 4	25	77	25	33.5	92.3	65	30.8	87.4	40.7
Building 5	28.5	83.3	24.5	34.5	94.1	68	31.3	88.3	44
Building 6	28	82.4	21.5	33.5	92.3	65	31.4	88.5	40.7
**January**									
Building 1	30	86	16	39	102.2	58.5	34.7	94.5	35.6
Building 2	28.5	83.3	28	34	93.2	65.5	32.3	90.1	48.2
Building 4	27.5	81.5	27.5	33.5	92.3	61.5	31.1	88	42.8
Building 5	30.5	86.9	31.5	34.5	94.1	91	32.3	90.1	65.9
Building 6	25	77	18	33	91.4	60	31.9	89.4	38.3
**May**									
Building 1	28	82.4	41	37	98.6	74	33.7	92.7	59.8
Building 2	28	82.4	43	36	96.8	65.5	34.6	94.3	58.1
Building 4	31	87.8	48	33	91.4	77.5	31.7	89.1	67.9
Building 5	26.5	79.7	47	38.5	101.3	100	31.6	88.9	77.3
Building 6	28.5	83.3	51	34.5	94.1	72	32	89.6	62.9
**August**									
Building 1	34	93.2	49.5	39	102.2	71	35.7	96.3	64.2
Building 2	31	87.8	72.5	37	98.6	97.5	34.7	94.5	87
Building 4	30.5	86.9	58.5	34	93.2	80	32.3	90.1	69.6
Building 5	31	87.8	55.5	35	95	81	32.8	91	71.5
Building 6	28.5	83.3	59	33.5	92.3	81	31.1	88	72.9

Note: Buildings with average temperatures above 32.2 °C (90 °F) are highlighted in red. Buildings 1 and 2 were repeatedly the warmest of the insect-rearing buildings.

**Table 2 vetsci-12-00191-t002:** AdDV positivity rates by age and season.

Season	2 Weeks	4 Weeks	6 Weeks	Totals
October	18/30 = 60%	13/21 = 61.9%	11/20 = 55%	42/71 = 59.2%
January	17/30 = 56.6%	21/25 = 84%	26/30 = 86.7%	64/85 = 75.3%
May	6/36 = 16.7%	7/30 = 23.3%	6/20 = 30%	19/86 = 22.1%
August	3/23 = 13%	10/30 = 33.3%	19/41 = 46.3%	32/94 = 34%
Totals	44/119 = 37%	51/106 = 48.1%	62/111 = 55.9%	157/336 = 46.7%

**Table 3 vetsci-12-00191-t003:** AdDV positivity rates by building and season.

Building	October	January	May	August	Totals
1	8/15 = 53%	15/20 = 75%	3/25 = 12%	7/26 = 26.9%	33/86 = 38.4%
2	6/10 = 60%	6/10 = 60%	9/15 = 60%	8/10 = 80%	29/45 = 64.4%
4	3/5 = 60%	10/10 = 100%	1/10 = 10%	3/15 = 20%	17/40 = 42.5%
5	2/9 = 22.2%	8/10 = 80%	5/25 = 20%	9/30 = 30%	24/74 = 32.4%
6	23/32 = 65.6%	25/35 = 71.4%	1/11 = 9.1%	5/13 = 38.5%	54/91 = 59.3%
Totals	42/71 = 59.2%	64/85 = 75.3%	19/86 = 22.1%	32/94 = 34%	157/336 = 46.7%

## Data Availability

All data can be provided by the corresponding author upon written or verbal request.

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
