# Peer review of "Characterizing the Roles of Life Stage and Season on the Prevalence of Select Viral Pathogens in Acheta domesticus Crickets on a Commercial Cricket Farm in the United States"

_vetsci, 2025, doi:10.3390/vetsci12030191_

Round 1
Reviewer 1 Report
Comments and Suggestions for Authors
The manuscript presents a detailed epidemiological analysis of the prevalence of three viruses in commercial Acheta domesticus populations. The study is relevant for the sustainable management of insect production and provides valuable insights into the impact of age and seasonality on the occurrence of viral infections. The methodological design is robust, employing modern qPCR techniques and appropriate statistical analysis. However, there are areas that could be improved to enhance the clarity and applicability of the findings. Additionally, expanding the discussion on the potential economic impacts of these infections in the cricket farming industry and preventive measures that could be implemented based on the results would be beneficial. The topic is pertinent to commercial insect production and industry biosafety, and the methodological approach is well-structured, with adequate sampling and coherent statistical analyses. The discussion comprehensively addresses environmental factors influencing the prevalence of the studied viruses, alongside the use of modern qPCR techniques for viral detection, which increases the reliability of the results. This study represents the first systematic approach to the epidemiology of these viruses in a large-scale cricket farming facility in the U.S.
Despite its relevance, the statistical data presentation could be improved to facilitate reader comprehension, suggesting the use of more explanatory tables and graphs highlighting comparisons between different periods and categories. The distinction between prevalence and viral load needs to be reinforced throughout the text to avoid confusion in data interpretation, and some paragraphs in the results section could be subdivided to improve readability. Although the manuscript raises hypotheses about viral transmission, it lacks more direct references to studies that have experimentally addressed horizontal and vertical transmission. It would be interesting to discuss more deeply the feasibility of aerosol transmission, as this factor could have direct implications for biosafety practices in commercial farms. Furthermore, expanding the discussion on possible mechanical and environmental vectors that could influence viral spread would be beneficial.
The manuscript cites several studies on Acheta domesticus viruses but could better explore comparisons with previous outbreaks and mitigation strategies already proposed in the literature. Including information on management practices already tested in affected farms could enhance the practical applicability of the findings, and some references are relatively outdated, making it advisable to include more recent studies to strengthen the theoretical foundation of the article. The justification for selecting sampling periods could be more detailed, explaining whether the chosen months were based on previous outbreak data. Considering that viral DNA detection does not necessarily indicate active infection, discussing whether other techniques, such as viral RNA quantification, could complement the findings would be useful. The description of negative and positive controls to ensure test accuracy could be more detailed, as well as an indication of whether sufficient biological replication was conducted to ensure statistical robustness in the qPCR experiments.
The manuscript presents a well-conducted and relevant study for the insect industry, providing valuable information on the epidemiological dynamics of viruses in Acheta domesticus. A review of specific aspects is recommended to improve the clarity of the results, contextualization with the literature, and discussion on viral transmission. With adjustments, the article has the potential to significantly contribute to the understanding of insect health management and the formulation of effective strategies for viral control and management in commercial farming operations.
Author Response
Reviewer #1 Comments and Suggestions:
Additionally, expanding the discussion on the potential economic impacts of these infections in the cricket farming industry and preventive measures that could be implemented based on the results would be beneficial.
Comments to reviewer #1: Thank you for the suggestion; however, the authors believe that discussing the potential economic impacts of these measures is beyond the scope of this study. Moreover, there are not any good objective/quantitative studies available to reference. A PubMed search using the key words cricket and economic and infections (1977-2025) revealed 10 papers, with none citing actual economic losses. Furthermore, the papers listed, similar to our paper, described the impact exists but needs to be measured. An additional search using only cricket and economic (1977-2025) revealed 111 papers, but again no information about economic losses beyond a simple statement. Our conclusion paragraph did already summarize the cost to the industry and discussed the need for future study ideas regarding environmental controls and housing controls (Lines 508-512): “Viral diseases have been linked to costly disease epidemics in captive-bred Acheta domesticus populations worldwide. The high mortality rates and decreased fecundity associated with these diseases have put a massive economic strain on the cricket industry and decreased our ability to integrate this sustainable animal protein source into more feed applications”. To further address your request we did add (lines 512-514): “To date, these economic costs have not been quantified but will be necessary to determine the extent of these losses and to reinforce the need for establishing preventive control measures to reduce these losses, similar to other livestock species”. Regarding preventative measures, we cited references 3, 31,32, and 33.
Despite its relevance, the statistical data presentation could be improved to facilitate reader comprehension, suggesting the use of more explanatory tables and graphs highlighting comparisons between different periods and categories.
Comments to reviewer #1: Thank you for the suggestion. Forest plots that display the odds ratios and their confidence intervals have been made for each logistic regression model. These were added to the supplementary materials (S1-S4) given the large number of graphs and figures already in use throughout the text.
The distinction between prevalence and viral load needs to be reinforced throughout the text to avoid confusion in data interpretation, and some paragraphs in the results section could be subdivided to improve readability.
Comments to reviewer #1: Thank you. The authors have added wording to improve the distinction between prevalence and viral load in lines 293-295 and lines 317-318: “Due to this, we were unable to perform univariate statistics or a binary logistic regression on positivity rates (infected: yes/no) as all crickets tested positive for the virus Therefore, an ordinal logistic regression model was constructed using the estimated viral load (how much virus was present in samples, 10n virus particles) as the dependent variable…”, and “…estimated viral load (number of virus particles, 10n)…”. A few of the paragraphs in the results section have also been further subdivided to improve readability (Line 243, 282, 305, 324, 330).
Although the manuscript raises hypotheses about viral transmission, it lacks more direct references to studies that have experimentally addressed horizontal and vertical transmission. It would be interesting to discuss more deeply the feasibility of aerosol transmission, as this factor could have direct implications for biosafety practices in commercial farms. Furthermore, expanding the discussion on possible mechanical and environmental vectors that could influence viral spread would be beneficial.
Comments to reviewer #1: We agree that there is a lack of references addressing the issue of horizontal and vertical transmission. This is because there are no studies that have addressed this concept yet for these three viruses and it is why we mention it as being a needed future study. The same goes for identifying whether these viruses can be transmitted through aerosolization. Short of the one article that mentioned finding virus particles in air filters (Szelei et al., 2011), no other studies have addressed transmission strategies other than fecal-oral. To confirm that we have not missed anything published recently, wet ran a PubMed search looking for additional studies that may have addressed transmission of these viruses using the key words and dates of AdDV, AdVVV, invertebrate iridovirus AND transmission OR aersolization, and 1975-2025, respectively. No additional studies could be found. The comments we make in the discussion suggesting a need for future transmission studies to be performed should suffice for now.
The manuscript cites several studies on Acheta domesticus viruses but could better explore comparisons with previous outbreaks and mitigation strategies already proposed in the literature. Including information on management practices already tested in affected farms could enhance the practical applicability of the findings, and some references are relatively outdated, making it advisable to include more recent studies to strengthen the theoretical foundation of the article.
Comments to reviewer #1: Unfortunately, there are no studies that have assessed mitigation strategies for these viruses. There are review papers that have discussed possible strategies, but none that have tested them so far. The authors intend to pursue mitigation strategies in future experiments, but there is nothing to comment on currently. We agree that some of the studies are outdated, unfortunately, these are the only scientific studies in the literature at this time as there was never a big push to continue studying them after their initial characterizations. Luckily, as the insects as feed and food industry has grown, so has interest in these viruses and more information should be coming out over the next few years.
The justification for selecting sampling periods could be more detailed, explaining whether the chosen months were based on previous outbreak data.
Comments to reviewer #1: The dates represent a month from each season (Fall, Winter, Spring, and Summer). The original dates were supposed to be more evenly distributed (every three months), however, Covid-19 delayed our spring sampling and left us with a larger gap between January 2020 and May 2020. We have included a sentence to indicate this delay in sampling (lines 131-132): “The delay in our Spring 2020 sampling period was due to Covid-19 restrictions.”
Considering that viral DNA detection does not necessarily indicate active infection, discussing whether other techniques, such as viral RNA quantification, could complement the findings would be useful.
Comments to reviewer #1: We have added a line of text stating that RNA quantification could have been used to confirm active infection, however, this was not deemed necessary given the extremely high prevalence rates found for our viruses (lines 495-498): “Additionally, the use of viral RNA quantification to indicate active infection could be used; however, given the extremely high prevalence rates in this study, which increased our confidence in calling these active infections, this line of testing was not pursued.”
The description of negative and positive controls to ensure test accuracy could be more detailed, as well as an indication of whether sufficient biological replication was conducted to ensure statistical robustness in the qPCR experiments.
Comments to reviewer#1: Lines 178-180 state that “Constructed plasmids that included the corresponding DNA sequences and nuclease-free water were used as positive controls and negative controls, respectively, for each virus.”. Is there something more specific you would like us to add? The authors believe that this is sufficient information, however, if the reviewer would like us to include the full plasmid construction and sequencing information to the supplementary file we are willing to do so. As for biological replicates, the use of triplicate replication is standard and should be sufficient to ensure statistical robustness for the purposes of this study. Following standard qPCR protocol, if the CT standard deviation was greater than 0.5 across the three replicates, the sample was re-run until the standard deviation across all three wells was within an acceptable range.

Reviewer 2 Report
Comments and Suggestions for Authors
House crickets, Acheta domesticus, have recently become an important feed for livestock species and humans. That makes it necessary to breed crickets on an industrial scale. However, the cricket industry has been plagued by virus outbreaks that can severely impact their production numbers.
Aim of the present study was to determine the prevalence of three common virus types in crickets under real world-large-scale industrial conditions. This is of high practical interest for cricket as well as livestock farmers.
The design of the experiments was well-considered, but also shows limitations. When temperature and humidity may affect virus outbreaks, a laboratory controlled study may have been more meaningful, but that's what the authors say too.
I have only minor suggestions for text improvement:
- authors should say in which development stage crickets were after 2, 4 and 6 weeks (tested age groups)
- Materials and Methods: give explanation for buffer used (AL, ATL); give g value for the centrifugation step
- explain numbers in Fig 1 A
- References: give species names correctly: Acheta domesticus, etc.; give correct abbreviation for PNAS (Proc. Natl. Acad. Sci. USA); give all species names in italics
Author Response
Reviewer #2 Comments and Suggestions:
I have only minor suggestions for text improvement:
- authors should say in which development stage crickets were after 2, 4 and 6 weeks (tested age groups)
Comments to Reviewer #2: Thank you for your suggestion. Crickets are often aged by week numbers; however, those numbers usually do not match up to “actual age” and really only correspond to the size of the cricket in inches. Instar number is not often used by producers until the last two instars when female/male determination can be made. Therefore, I have included the size of the crickets in addition to the week ages to allow for better clarity and reference to other studies (lines 119-120): Three different life stages of crickets were sampled: 2-week (nymphs 1/4” in size), 4-week (nymphs 1/2” in size), and 6-week-olds (adults 1” in size)..
- Materials and Methods: give explanation for buffer used (AL, ATL); give g value for the centrifugation step
Comments to Reviewer #2: Buffer AL and ATL are proprietary buffers that come with the kit used for DNA extraction (Qiagen DNeasy Blood and Tissue kit). The g values are provided in the kit instructions and were not changed for our experiments. There are three different centrifugation steps between wash cycles at different rates, so it becomes bulky to add that much detail for something that is already published by the manufacturer.
- explain numbers in Fig 1 A
Comments to Reviewer #2: Thank you for the comment. The numbers were data points that were outliers and only corresponded to the chronological list number of the data. I have edited the figure to include the outlier circles but have removed the corresponding numbers so as to not confuse readers.
- References: give species names correctly: Acheta domesticus, etc.; give correct abbreviation for PNAS (Proc. Natl. Acad. Sci. USA); give all species names in italics
Comments to Reviewer #2: All formatting issues have been changed as requested.

Round 2
Reviewer 1 Report
Comments and Suggestions for Authors.